# Stabilization of three-dimensional charge order in $YBa_2Cu_3O_{6+x}$ via epitaxial growth

M. Bluschke[1,2], A. Frano[3,4,5], E. Schierle[2], D. Putzky[1], F. Ghorbani[1], R. Ortiz[1], H. Suzuki[1], G. Christiani[1], G. Logvenov[1], E. Weschke[2], R.J. Birgeneau[3], E.H. da Silva Neto[6], M. Minola[1], S. Blanco-Canosa[7,8,9] & B. Keimer[1]

Incommensurate charge order (CO) has been identified as the leading competitor of high-temperature superconductivity in all major families of layered copper oxides, but the perplexing variety of CO states in different cuprates has confounded investigations of its impact on the transport and thermodynamic properties. The three-dimensional (3D) CO observed in $YBa_2Cu_3O_{6+x}$ in high magnetic fields is of particular interest, because quantum transport measurements have revealed detailed information about the corresponding Fermi surface. Here we use resonant X-ray scattering to demonstrate 3D-CO in underdoped $YBa_2Cu_3O_{6+x}$ films grown epitaxially on $SrTiO_3$ in the absence of magnetic fields. The resonance profiles indicate that Cu sites in the charge-reservoir layers participate in the CO state, and thus efficiently transmit CO correlations between adjacent $CuO_2$ bilayer units. The results offer fresh perspectives for experiments elucidating the influence of 3D-CO on the electronic properties of cuprates without the need to apply high magnetic fields.

[1] Max Planck Institute for Solid State Research, Heisenbergstr. 1, 70569 Stuttgart, Germany. [2] Helmholtz-Zentrum Berlin für Materialien und Energie, Wilhelm-Conrad-Röntgen-Campus BESSY II, Albert-Einstein-Str. 15, 12489 Berlin, Germany. [3] Department of Physics, University of California, Berkeley, California 94720, USA. [4] Advanced Light Source, Lawrence Berkeley National Laboratory, Berkeley, California 94720, USA. [5] Department of Physics, University of California San Diego, La Jolla, California 92093, USA. [6] Department of Physics, University of California, Davis, California 95616, USA. [7] CIC nanoGUNE, 20018 Donostia - San Sebastian, Basque Country, Spain. [8] IKERBASQUE, Basque Foundation for Science, 48011 Bilbao, Basque Country, Spain. [9] Donostia International Physics Center, 20018 Donostia - San Sebastian, Basque Country, Spain. Correspondence and requests for materials should be addressed to S.B-C. (email: S.Blanco@nanogune.eu) or to B.K. (email: B.Keimer@fkf.mpg.de)

Recent nuclear magnetic resonance and X-ray scattering experiments have identified incommensurate charge order (CO) as a generic instability of copper oxide superconductors[1–5]. The competition of CO and high-temperature superconductivity was identified as a key factor shaping the phase diagram in the underdoped regime. These results have stimulated a profusion of research on the influence of collective charge fluctuations on the mechanism of superconductivity and on the anomalous transport and thermodynamic properties in the normal state of the underdoped cuprates[6–9]. However, many aspects of charge ordering differ considerably between various families of cuprates. Specific CO states reported so far include two-dimensionally (2D) short-range ordered states in hole-doped Bi- and Hg-based materials[3,4] and electron-doped $Nd_{2-x}Ce_xCuO_4$[5,10], longer-range quasi-2D CO with different doping-dependent incommensurabilities in $La_{2-x}Sr_xCuO_4$[11–13] and $YBa_2Cu_3O_{6+x}$[1,2,14] (YBCO), and three-dimensional (3D) CO in YBCO in high magnetic fields[15–17]. These differences reflect the distinct lattice architectures and doping mechanisms of individual compound families that are difficult to capture in quantum many-body theories. The generic description of the interplay between CO and superconductivity thus requires a model system where CO correlations can develop with minimal disruption by doping-induced disorder.

Over the past decade, this role has been served by the YBCO system where resistivity, magnetization, and specific heat experiments in magnetic fields high enough to substantially degrade superconductivity have revealed pronounced quantum oscillations even in the underdoped regime[9,18–21]. The oscillation period inferred from quantum transport data implies a small electron pocket, and thus indicates a reconstruction of the hole-like Fermi surface due to translational symmetry breaking. The size of the experimentally observed electron pocket is consistent with the one expected from CO-induced folding of the Fermi surface[22]. The experimental discovery of 3D-CO with in-plane correlation lengths comparable to the cyclotron radius of the electron orbit in high magnetic fields has provided further support for models relating the fermiology of YBCO to CO. However, several important issues remain unresolved, including especially the lack of evidence for hole pockets (the counterpart of electron pockets in the reconstructed Fermi surface) in quantum oscillation data. At the same time, the need to impose high magnetic fields precludes experiments (such as angle-resolved photoemission) that could reveal the full Fermi surface geometry in the 3D-CO state.

Here we report a resonant X-ray scattering (RXS) study of moderately doped epitaxial thin films of YBCO, which has revealed Bragg reflections indicative of 3D-CO in zero magnetic field. The high-field 3D-CO state in bulk YBCO has thus far only been investigated by non-resonant hard X-ray diffraction, which probes the spatial distribution of the entire charge density. RXS with soft X-rays tuned to the Cu $L_{3,2}$-absorption edge, on the other hand, is a spectroscopic measurement, which selectively probes spatial variations of the electronic states near the Fermi level[6,23]. Based on spectroscopic information, RXS can also distinguish contributions of different crystallographic sites to charge-ordered superstructures[23]. However, the technical challenges associated with producing high magnetic fields have thus far prohibited RXS studies of the 3D-CO state in bulk YBCO. Our thin-film data now demonstrate that Cu atoms in the charge-reservoir layers between the $CuO_2$ sheets participate in the CO state. The RXS results thus yield fresh insight into the mechanisms stabilizing the unique 3D-CO state in YBCO, and point out new opportunities to explore the influence of CO on the transport and thermodynamic properties of the cuprates in the absence of high magnetic fields.

## Results

**Lattice structure.** We investigated underdoped YBCO films of different thicknesses that were grown epitaxially on $SrTiO_3$ (STO) substrates using pulsed laser deposition (see "Methods" and Table 1). The in-plane lattice parameters of the nearly cubic perovskite structure of STO are $a = b = 3.905$ Å, larger than those in the orthorhombic structure of bulk YBCO ($a = 3.83$ Å and $b = 3.88$ Å). STO thus applies tensile strain at least to the first layers deposited in the growth process. Epitaxial strain generally relaxes along the growth direction, and in our 50 nm thick films the volume-averaged in-plane lattice constants observed were $a = b = 3.86(2)$ Å. A more profound difference between the crystal structure of our YBCO films and that of bulk YBCO is indicated by the absence of any of the conventional CuO chain ordering patterns[24] associated with the charge-reservoir layer in our scattering experiments. The local orthorhombic distortion of the unit cell caused by the dopant oxygens in the charge-reservoir layer is therefore uncorrelated, and the crystal structure is essentially tetragonal to any non-local probe. This change in lattice symmetry is presumably a consequence of the epitaxial relationship with the STO substrate, which enforces a tetragonal structure in the deposited layers; prior work has shown that such effects can persist over tens of nm in the growth direction[25,26]. We will discuss the influence of this distinct crystal symmetry on the CO pattern below.

**2D and 3D charge-order correlations.** Reciprocal space mapping of epitaxial YBCO films performed with resonant X-rays tuned to the Cu $L_3$-absorption edge reveals a rod of diffracted intensity in the $H−L$ plane peaked at incommensurate in-plane momentum transfer (Fig. 1), corresponding to the well-studied 2D-CO in the $CuO_2$ planes of bulk underdoped YBCO[2,27]. (The reciprocal-

### Table 1 Fitting of the 3D-CO peaks

| $T_c$ (K) | Thickness (nm) | $q_{//}$ (r.l.u.) | Peak area from $H$-scan (arb. units) | Peak area from $L$-scan (arb. units) | Correlation length along a (nm) | Correlation length along c (nm) |
|---|---|---|---|---|---|---|
| 41 | 50 | — | — | — | — | — |
| 50 | 50 | −0.329(1) | 0.7(1) | 2.2(2) | 8(1) | 6(1) |
| 53 | 50 | −0.329(1) | 1.7(2) | 10(1) | 10(1) | 6(1) |
| 55 | 50 | −0.329(1) | 0.9(1) | 4.3(4) | 8(1) | 5(1) |
| 62 | 50 | — | — | — | — | — |
| 53 | 20 | −0.324(1) | 0.7(2) | 0.8(2) | 3(1) | 3(1) |

Fit parameters for Lorentzian fits to the 3D-CO peak measured at 12 K in a series of YBCO thin films grown on STO (100). The raw data for the 50 nm films is shown in Fig. 2e, f and for the 20 nm film in Fig. 1c. Since the 3D-CO is still observable in the 50 nm films at 300 K, fluorescence backgrounds of the type shown in Fig. 2c, d, f (dashed lines) were subtracted from the 12 K data before fitting. In contrast the 3D-CO reflection in the 20 nm film is broader, making necessary the subtraction of the 300 K data (Fig. 1c, red points) before fitting. Errors in the tabulated values were estimated from a sensitivity analysis of the various background subtraction procedures

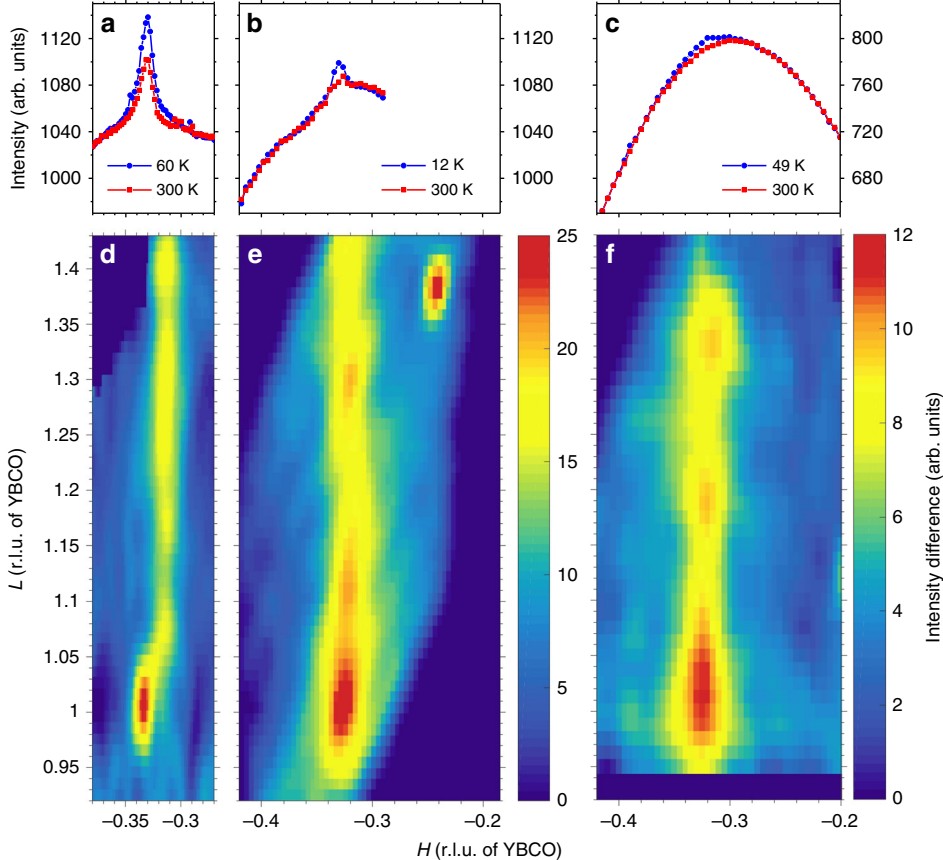

**Fig. 1** Reciprocal space mapping. Three YBCO films grown on STO (001) are studied; **a**, **d** 50 nm, $T_c = 53$ K; **b**, **e** 50 nm, $T_c = 55$ K; **c**, **f** 20 nm, $T_c = 53$ K. Scattering data were collected using $\sigma$ polarized photons tuned to the Cu L$_3$-absorption edge. Panels **a–c** show the high and low temperature scattering intensity along the reciprocal space direction (H 0 1). Panels **d–f** exhibit reciprocal space maps in the $H-L$ plane, these are produced by subtracting the scattered intensity detected at 300 K from the equivalent map measured at low temperature. The color maps were smoothed using a Gaussian filter

space coordinates $H$, $K$, $L$ are quoted in reciprocal lattice units, r.l.u., based on the tetragonal structure of the YBCO films.) The diffraction rod extending in the $L$ direction indicates 2D short-range order without correlation in the crystallographic **c** direction, qualitatively analogous to bulk YBCO in the absence of magnetic fields. The in-plane correlation lengths are comparable to those previously observed in bulk YBCO[27], indicating that the absence of oxygen order in the films does not have a major impact on the 2D-CO, in agreement with prior work on oxygen-disordered bulk YBCO[28].

Our main finding is the discovery of an additional temperature dependent diffraction peak centered at momentum transfer $\mathbf{q}_{3D} = (-0.329\ 0\ 1)$ in multiple YBCO films (Figs. 1 and 2e). The reflection intensity is sharply peaked in all three reciprocal-space directions, indicating 3D phase coherence, and is also present for positive in-plane momentum transfer $\mathbf{q} = (0.329\ 0\ 1)$. While it is clearly observable for incident photons with $\sigma$ polarization (electric field $\boldsymbol{\varepsilon} \| b$), the peak at $\mathbf{q}_{3D}$ is completely suppressed with $\pi$ polarized X-rays (Fig. 3f). Since magnetic scattering is typically stronger for incident $\pi$ polarized X-rays, vanishing intensity of the 3D diffraction feature for incident $\pi$ polarization ($I_\pi/I_\sigma = 0$), leads us to conclude a non-magnetic origin and attribute the resonant diffraction peak at $\mathbf{q}_{3D}$ to 3D-CO. Moreover the dramatic polarization dependence is not unexpected. In bulk YBa$_2$Cu$_3$O$_{6.67}$ at zero field, the ratio of scattering intensities $I_\pi/I_\sigma$ recorded at various $L$ values along the 2D-CO rod is observed to vanish as the momentum transfer along **c** approaches $L = 1$[29]. Considering all of these factors, we

identify the 3D-CO reflection observed in thin films of YBCO as analogous to the behavior of bulk YBCO in high magnetic fields.

**XAS and RXS energy profiles**. RXS further provides spectroscopic information about the 3D-CO via analysis of the photon energy dependence. The spectral profile of the 3D-CO reflection near the Cu L$_3$ edge is plotted in Fig. 3a and compared with X-ray absorption spectroscopy (XAS) of the same film, measured with X-ray polarization parallel to the planar Cu–O bonds. Both the RXS and XAS processes involve the promotion of a core $2p$ electron into an unoccupied electronic state near the Fermi level. Since the XAS technique does not provide spatially resolved information, the measured lineshape represents the sum over all absorption processes associated with all ions in the material. In the YBCO structure there are two distinct Cu sites: Cu(I) located in the charge-reservoir layer containing the CuO chains, and Cu(II) in the CuO$_2$ planes. As a function of oxygen doping the Cu(I) sites evolve from a $3d^{10}$ ground-state configuration, associated with empty CuO chains, to a $3d^9 + 3d^9\underline{L}$ configuration associated with filled chains[30]. Here $\underline{L}$ indicates a hole on the ligand oxygens. Since the dopant oxygen ions are intercalated into the charge reservoir layer and not into the CuO$_2$ planes, the Cu(II) sites receive a comparatively light hole doping and evolve between the $3d^9$ and $3d^9 + 3d^9\underline{L}$ states as a function of increasing oxygen content. While the main maximum of the Cu L$_3$ XAS can be associated with transitions to the $3d^9$ state of the Cu(II) sites, transitions to the $3d^9\underline{L}$ and $3d^{10}$ states of the Cu(I) sites occur

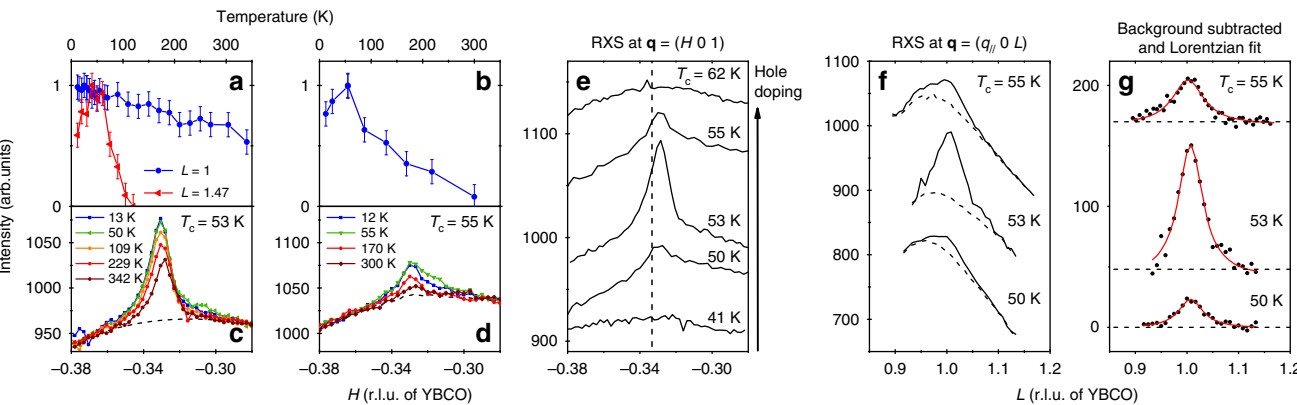

**Fig. 2** Temperature and doping dependence. **a–d** Temperature dependence of the scattering intensity along the $(H\ 0\ 1)$ reciprocal space direction for two 50 nm films with $T_c = 53$ K (**a**, **c**) and 55 K (**b**, **d**). The backgrounds (dashed lines) in **c**, **d** were achieved by slightly detuning the detector angle away from the Bragg condition, and then subtracted before integrating to produce the normalized data points in **a**, **b** (blue circles). For comparison the temperature dependent intensity of the 2D-CO rod collected at $L = 1.47$ is shown in **a** (red triangles). **e** Scattering intensity at 12 K along $(H\ 0\ 1)$ for five YBCO films of equal thickness (50 nm), but varying doping, as indicated by variation in the superconducting $T_c$ (see "Methods"). The commensurate momentum transfer $H = -\frac{1}{3}$ is indicated by the dashed vertical line. **f** $L$-scan of the 3D-CO reflection for the 50 nm films with $T_c = 50$, 53, and 55 K. The background was obtained in the same manner as described for **c**, **d**. **g** Same data as in **f** after background subtraction (black circles), and fitting with Lorentzian profiles (solid red line). **a–g** All data were collected using $\sigma$ polarized X-rays tuned to the Cu $L_3$ resonance. For clarity, vertical offsets were added to the curves in **e–g**. Although our data does contain statistical noise, the error in our analysis is dominated by the uncertainty in determining the intrinsic background signal. Accordingly, the error bars in **a**, **b** were estimated from a sensitivity analysis of the background subtraction procedure

for slightly higher photon energies, 932.7 and 934.3 eV, respectively[30]. The latter of these is clearly visible in the XAS profile, whereas the transition associated with the $3d^9\underline{L}$ state is hardly recognizable. Inspection of the 3D-CO resonance profile reveals a main maximum coinciding with the XAS maximum, a strong shoulder around 933 eV, and a second significantly weaker shoulder at even higher energy, corresponding to the Cu(I) $3d^{10}$ states. The enhancement of the resonant scattering intensity at energies associated with transitions to states of both the Cu(I) and Cu(II) ions indicates the participation of both Cu sites in the 3D-CO.

For comparison, we have measured the resonance profiles of a number of other Bragg peaks associated with various ordering tendencies intrinsic to underdoped YBCO. Figure 3b shows the resonance profile of the 2D-CO measured in one of our YBCO films. In agreement with previous studies[31], the 2D-CO reflection resonates solely at the energy corresponding to transitions into the $3d^9$ states of the Cu(II) ions. In Fig. 3c we display the resonant enhancement of the (001) structural Bragg peak measured in the same film. The same XAS spectrum is plotted in Fig. 3a–c to facilitate comparison with the resonant scattering profiles.

Finally, Fig. 3d shows the resonant enhancement of an ortho-II oxygen superstructure peak measured in bulk single crystal YBCO$_{6.55}$, along with XAS measured in the same crystal with $\varepsilon||b$. Although the temperature dependence of the 3D-CO peak is inconsistent with that of a structural Bragg reflection (discussed below), one might be tempted to suggest that this peak originates from an ordered oxygen superstructure in the charge reservoir layer. In order to exclude this possibility, we consider the polarization-dependent resonant enhancement of the ortho-II oxygen superstructure peak displayed in Fig. 3d, which is consistent with previous resonant scattering studies of ortho-II, -III, and -VIII ordered YBCO[30,31]. Since the dopant oxygens order into Cu–O chains in the charge reservoir layer, it is expected that this superstructure modulates the atomic scattering form factors of the Cu(I) ions. Accordingly the resonance profile of the ortho-II superstructure reflection is peaked at photon energies associated with the $3d^{10}$ and $3d^9\underline{L}$ states of the Cu(I) ions. The fact that the ortho-II peak does not resonate at the energy of the XAS maximum (associated with the Cu(II) sites),

together with the polarization dependence, which actually demonstrates stronger scattering from the Cu–O chain with $\varepsilon||c$, leads us to conclude that the 3D-CO observed in our YBCO films does not correspond to a Cu–O chain ordering superstructure.

In contrast to both the (001) Bragg peak and the ortho-II superstructure peak, neither the 2D-CO nor the 3D-CO were detected at the Cu $L_2$ resonance (951.4 eV). The 3D-CO peak intensity recorded at the Cu $L_3$ resonance is at least eight times greater than the noise level, from which we derive a lower limit on the $L_3/L_2$ branching ratio of at least 8. A similar lower limit on the $L_3/L_2$ branching ratio can be extracted from RXS measurements of the 2D charge correlations in bulk YBCO[27]. This lower limit may help to distinguish the nature of the ordering type when modeling the resonant scattering profile across the Cu $L_{3,2}$ edges. Additionally this lower limit further discriminates the 2D- and 3D-CO reflections from the (001) Bragg peak and ortho-II superstructure peak, which both exhibit $L_3/L_2$ branching ratios of less than 8.

**Temperature dependence**. The intensity of the 2D-CO rod evolves upon cooling as expected for bulk YBCO, onsetting below ~120 K and achieving a maximum intensity at the superconducting $T_c$ (red triangles in Fig. 2a for the 50 nm film with $T_c = 53$ K). In stark contrast, the 3D-CO peak observed at $\mathbf{q}_{3D}$ exhibits a monotonic intensity increase upon cooling, and no saturation (blue circles in Fig. 2a, and raw data in Fig. 2c). Although a significant intensity remains at room temperature, the intensity variation between 12 K and 300 K is too strong to be accounted for by the Debye-Waller factor, and therefore makes a purely structural origin of this reflection unlikely. In fact, the greater thermal stability of the 3D-CO compared to the 2D-CO may be a simple consequence of the increased dimensionality of the CO correlations. The temperature dependence of the 3D-CO peak in a second film with $T_c = 55$ K, shown in Fig. 2b, d is much more pronounced with almost no intensity remaining at room temperature. Additionally, its temperature dependence demonstrates an apparent maximum at $T_c$, however this maximum likely arises from contamination of the $H$-scan with intensity

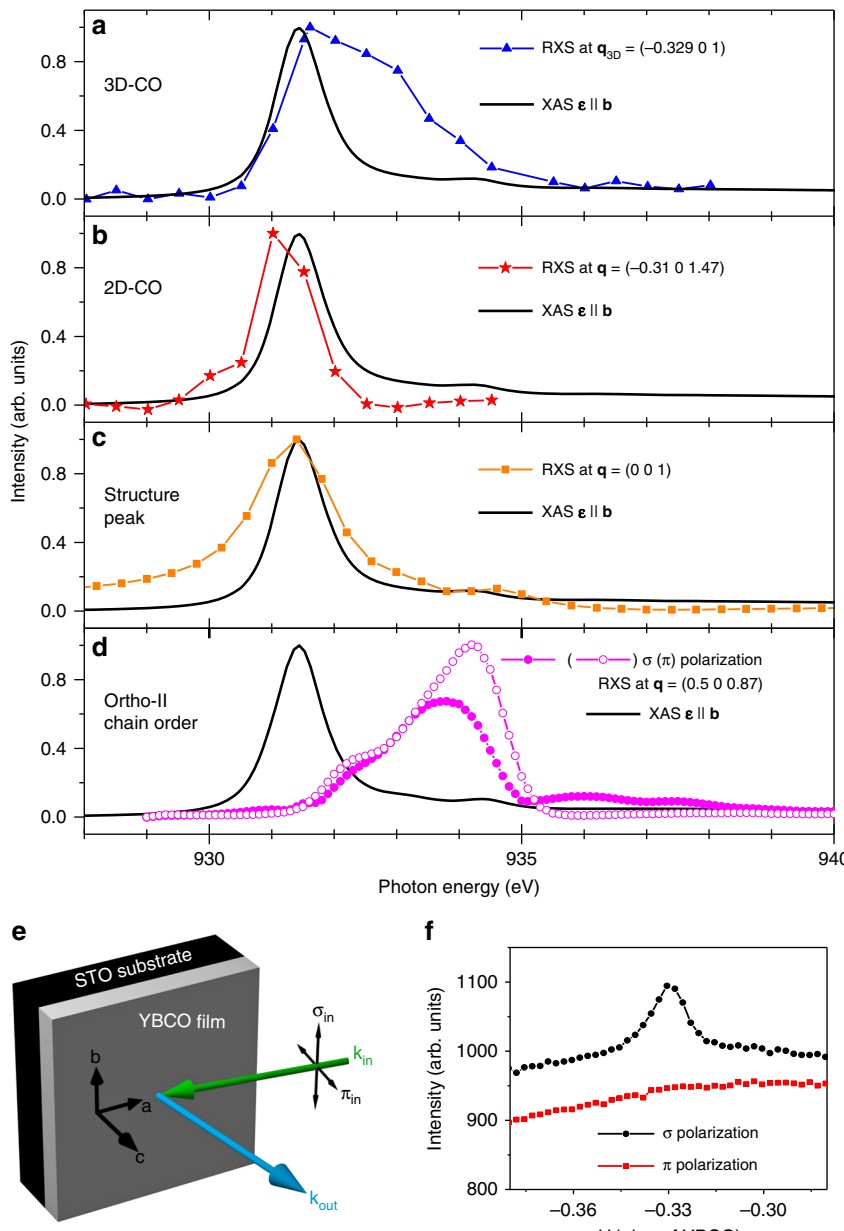

**Fig. 3** Photon energy and polarization dependence. **a–d** Comparison of various energy dependent scattering intensities across the Cu L$_3$ edge. The data in **a–c** were measured on a 50 nm YBCO thin film grown on SrTiO$_3$ (no chain order), whereas the data in **d** were collected from a single crystal of ortho-II ordered YB$_2$Cu$_3$O$_{6.55}$. **e** Scattering geometry used to study the 3D-CO reflection at **q**$_{3D}$. **f** Polarization dependence of the 3D-CO reflection as measured in **e**, and with photon energy tuned to the Cu L$_3$ edge

from the CO rod which in this film occurs at approximately the same in-plane momentum transfer as the 3D-CO and with comparable intensity. Unlike the integrated intensity, the correlation length is already saturated at 300 K and found to be between 8 and 10 nm along **a** and between 5 and 6 nm along **c**. Among the series of samples studied, only the 20 nm film shows significant deviation in the correlation lengths of the 3D-CO (see Table 1).

**Film thickness dependence.** In order to test the extent to which CO correlations are affected by tensile strain imposed by the STO substrate, which expands the in-plane lattice parameters of YBCO, we have prepared a thinner 20 nm YBCO film with $T_c = 53$ K. Since epitaxial strain is expected to relax over the first 10–20 nm in the growth direction, a purely strain-induced

mechanism for 3D phase coherence would exist only in the near-substrate region. In this case, reducing the film thickness from 50 to 20 nm (compare Fig. 1a, d with Fig. 1c, f), should leave the absolute intensity of the 3D-CO unchanged. In fact, however, both the intensity and correlation length of the 3D-CO reflection were found to decrease significantly in the 20 nm thin film (see Table 1). We conclude that the 3D-CO exists throughout the entire thickness of the 50 nm films, and that confining the 3D-CO by reducing the film thickness causes a decrease in its correlation length.

**c-axis correlation and a-b anisotropy.** The in-plane and out-of-plane correlation lengths associated with the 3D-CO were extracted from the H- and L-scans shown in Fig. 2e–g, and are listed in Table 1. They are comparable to those in the field-induced

3D-CO in bulk ortho-II and ortho-VIII ordered YBCO[15–17]. In contrast, the 2D-CO rods observed in our YBCO films exhibit no enhancement of the scattering intensity towards $L = 1.5$ (Fig. 1), indicating that the antiphase correlation between 2D-CO correlations in neighboring $CuO_2$ planes, previously established for bulk underdoped YBCO[16,32], is not present in our films. While one might be tempted to suggest that the 2D-CO rods observed in Fig. 1 are simply tails of the 3D-CO peak at $L = 1$, the in-plane separation of the rod and the 3D charge peak in Fig. 1d disfavors this interpretation. A more feasible scenario to explain the observed diffraction pattern involves a microscopic phase separation of regions hosting a 2D-CO (rod) and regions hosting a 3D-CO (peak at $L = 1$). Alternatively, the 2D and 3D-CO may coexist spatially, but propagate along distinct in-plane crystallographic directions. In this scenario both reciprocal-space features appear along $H$ in Fig. 1 due to the presence of spatially separated domains in which the **a** and **b** directions are exchanged. Due to the effective tetragonality of the crystal structure, a scenario involving distinct charge ordering instabilities along distinct in-plane crystallographic directions would imply a nematic spontaneous symmetry breaking. Hard X-ray diffraction studies[16] of bulk single crystal YBCO in static magnetic fields reveal the abrupt emergence of a sharp reflection at $\mathbf{q} = (0\ K_{CO}\ 1)$ above 15 T, indicating a field-induced 3D phase coherence of the CO propagating along the **b** direction, whereas along **a** the CO remains two dimensional. However, in ortho-ordered single crystal YBCO, the equivalence of the **a** and **b** directions is already broken by the CuO chain order in the charge-reservoir layers.

## Discussion

Our RXS study has demonstrated 3D-CO in YBCO thin films in zero magnetic field and uncovered spectroscopic evidence for the participation of Cu(I) sites, located in the charge reservoir layer, in the CO state. The electronically active charge reservoir distinguishes YBCO from most other cuprates, where the dopant ions have closed electronic shells. Its prior manifestations include a uniaxial in-plane anisotropy of the transport properties due to the contributions of partially occupied electronic states in the CuO chains to the Fermi surface. The RXS energy profile of Fig. 3a now reveals a spatial modulation of the valence electron density in the charge-reservoir layer with the same incommensurate wave vector that characterizes the 2D-CO in the $CuO_2$ layers. The charge-modulated CuO chain layer can thus effectively transmit CO correlation between adjacent $CuO_2$ bilayers, bridging the large distance between $CuO_2$ layers that characterizes most other cuprate families. The phenomenology of the 3D-CO state in the YBCO films shares key features with the one previously observed by non-RXS in bulk YBCO in high magnetic fields[16], including particularly its coexistence with the 2D-CO state with the same incommensurability. It is therefore highly likely that the charge modulation of Cu chain states directly observed in our thin-film data also pertains to 3D-CO in bulk YBCO, where RXS experiments in high magnetic fields have thus far not proven feasible. Our results therefore go a long way towards explaining why 3D-CO has so far only been observed in YBCO, and not in the other cuprate families.

Nonetheless, there are significant differences between the 3D-CO state in bulk YBCO and the one we have reported in YBCO films. Whereas the former state is only realized in high magnetic fields and temperatures below ~60 K, the latter state is stable in zero field and temperatures up to at least 300 K. Based on the observed dependence of CO correlations on film thickness, it is unlikely that tensile strain induced by the STO substrate is primarily responsible for the unexpected robustness of the 3D-CO state in the films. Likewise, pinning to lattice defects such

as stacking faults and dislocations generated by steps on the substrate surface is unlikely to play a central role in generating the 3D-ordered state, because the corresponding Bragg reflections are consistently observed in multiple films, with in-plane correlation lengths of order 8 nm (Table 1). However, we cannot rule out that extended defects may promote phase separation into regions with and without 3D order. The apparent lack of competition between 3D-CO and superconductivity in our YBCO films may support the notion of mesoscopic phase separation between regions hosting 3D-CO, where superconductivity is suppressed, and regions hosting competing 2D-CO and superconducting states. Determining whether the 2D-CO and 3D-CO states coexist locally requires the application of a spatially resolved technique capable of imaging CO domains. The resolution of this question is thus left for future studies.

The most salient difference between the lattice structures of bulk and thin-film YBCO is the absence of ortho ordering of the dopant oxygen ions in the films, such that their lattice symmetry is tetragonal. The difference in lattice symmetry, which is presumably enforced by epitaxy with the STO substrate, can have a substantial influence on the electronic structure. The CuO chains in bulk YBCO give rise to a separate Fermi surface sheet and cause a complex folding of the $CuO_2$ plane-derived sheets, which degrade the nesting properties of the Fermi surface. In contrast, the Fermi surface of the effectively tetragonal structure of the films is expected to be more strongly nested, albeit broadened by disorder due to the randomly distributed oxygen dopants. The enhanced nesting, with nearly parallel Fermi surface segments in the Cu–O bond direction, is expected to favor a charge density wave instability. Detailed electronic-structure calculations are required to assess whether this effect is sufficient to explain the strongly enhanced 3D-CO instability we observed. The absence of 3D-CO in both films and bulk crystals of highly underdoped, quasi-tetragonal YBCO is reminiscent of the reduced dimensionality of superconductivity previously reported for highly underdoped YBCO films[33]. Both effects may be consequences of the closed shell $3d^{10}$ electronic configuration characterizing the Cu(I) ions in the underdoped limit, which closes the Cu(II)–O–Cu(I)–O–Cu(II) conduction pathway along the **c** direction and effectively constrains the electronic correlations to the $CuO_2$ planes, similar to the closed-shell cations $Hg^{2+}$ and $La^{3+}$ in $HgBa_2CuO_{4+\delta}$ and in the La-based cuprates.

Alternatively, one might consider scenarios in which charge transfer[34], orbital hybridization[35], or phonon hybridization[36] at the YBCO-STO interface nucleate the 3D-CO instability, which then propagates throughout the entire film. A related long-range electronic proximity effect has recently been reported for 2D-CO at $(La,Ca)MnO_3$–YBCO interfaces[37]. We also note that incommensurate 3D-CO with comparable ordering temperatures is observed in insulating layered nickelates of composition $R_{2-x}Sr_xNiO_4$ (where $R$ is a rare-earth ion)[38], where $NiO_6$ octahedral tilts and rotations are believed to facilitate the coupling between electronic charge order and the lattice structure. However, analogous soft structural degrees of freedom are not apparent in the YBCO structure.

We end our discussion by pointing out parallels to other recent observations on cuprates. In particular, recent RXS experiments on overdoped bulk $(Bi,Pb)_{2.12}Sr_{1.88}CuO_{6+\delta}$ revealed 2D-CO diffraction features with large in-plane correlation lengths that persist up to at least room temperature[39]. These findings were discussed in terms of enhanced nesting properties of the Fermi surface in proximity to the Lifshitz point, where a 2D van Hove singularity crosses the Fermi level. Together with our results, these observations illustrate the delicate stability conditions for CO in the cuprates, and offer new targets for theoretical research aimed at a realistic description of the phase behavior of these materials.

Independent of its microscopic origin, the 3D-CO state we have discovered in epitaxial YBCO films is much more accessible to experimental probes than the high-field state in bulk YBCO, and thus provides fresh perspectives for experimental exploration. To assess the influence of 3D-CO on the Fermi surface and macroscopic properties, it will be particularly interesting to perform angle-resolved photoemission and transport experiments on YBCO films and compare the results with those for bulk YBCO.

## Methods

**Thin film growth and characterization.** The YBCO thin films were grown by pulsed laser deposition on (100) oriented $SrTiO_3$ substrates at the Max-Planck-Institute for Solid State Research in Stuttgart. A KrF excimer laser was used to ablate material from a commercially purchased stoichiometric $YBa_2Cu_3O_7$ target, with a 2 Hz pulse rate and an energy density of $1.6\,J\,cm^{-2}$. The STO substrate was kept under an oxygen partial pressure of 0.5 mbar and held at a temperature of 730 °C during the deposition process. This procedure yields (001) oriented films of YBCO with in-plane twinning. Following the deposition process a series of six films were annealed for 15–60 min at 500 °C in oxygen partial pressures ranging between 0.08 and 3 mbar, resulting in a series of superconducting critical temperatures (see Table 1). Whereas in bulk YBCO the hole doping level $p$ can be determined by measuring the out-of-plane lattice parameter[40], the tensile-strain-induced reduction of the $c$ lattice spacing precludes a quantitative evaluation of the doping using this method. Similarly, a universal relationship between hole doping and $T_c$ is difficult to establish for thin-film YBCO, as the critical temperature depends sensitively on the substrate, strain, and growth conditions. Accordingly, we have chosen to study a series of YBCO films all grown on STO substrates with constant growth parameters. Although we are unable to accurately determine the hole doping in the $CuO_2$ planes, we attribute relative variations in $T_c$ among our systematically grown films to changes in the hole doping achieved during the annealing procedure. Furthermore, we note that the $c$ lattice parameters of our films were all found to be within the range 11.69–11.73 Å, consistent with bulk underdoped YBCO.

The superconducting $T_c$ was determined using a mutual inductance device, and defined as the onset temperature of the Meissner effect. Electrical transport measurements were performed using a Quantum Design Physical Properties Measurement System (PPMS), indicating $T_c$ consistent with those determined from mutual inductance measurements, and revealing residual resistivities as low as 50 μΩcm. Transmission electron microscopy of one of the same YBCO films studied here can be found in the supplementary information of Ref.[37].

**Resonant soft X-ray absorption and scattering.** The RXS experiments were performed using linearly polarized photons produced by the elliptical undulator of the UE46-PGM1 beamline at the BESSY II synchrotron at the Helmholtz-Zentrum-Berlin. The sample environment consists of an ultra-high-vacuum two-circle diffractometer in which the sample is mounted directly to a liquid Helium flow cryostat. Careful adjustment of the sample position with respect to the X-ray beam was accomplished to a precision of ~5 μm enabling reproducible background intensities when cycling the temperature. This precision is necessary in order to accurately detect charge reflections with signals as weak as 1% of the background intensity. Scattered photons were detected using a standard photodiode, and the reciprocal space maps were produced by scanning the angle of the sample with respect to the incident beam for a series of detector angles. The photodiode does not discriminate photon energies, implying that the measured intensities represent an integration over all elastic and inelastic scattering processes.

The CO resonant scattering profiles measured on a 50 nm YBCO film and plotted in Fig. 3a, b were produced by performing $H$-scans ($\theta$-scans) through the 3D-CO peak (2D-CO rod) for a series of incident photon energies around the Cu $L_3$ edge. Each of these scans was then analyzed by subtracting a second-order polynomial fit to the background, and either numerically integrating the remaining peak, or fitting a Lorentzian to it. Both methods yield comparable lineshapes. In contrast, the (001) Bragg peak from a 50 nm YBCO film (Fig. 3c), and the ortho-II chain order peak from a $YBa_2Cu_3O_{6.55}$ bulk single crystal (Fig. 3d), were both strong enough to allow constant-**q** energy scans, which simply measure the scattered intensity directly at the peak maximum as a function of incident photon energy. For each peak, the constant-**q** energy scan was performed three times: once directly on the peak maximum, and twice for positions just off the peak, achieved by detuning the $2\theta$ angle an equal amount above and below the Bragg condition. The two off-peak constant-**q** scans were then averaged and subtracted from the on-peak scan to remove contributions to the resonance profile associated with the energy dependent fluorescence background. The XAS spectra were measured at the same endstation, and collected in the total electron yield mode.

**Data availability.** The data that support the findings of this study are available from the corresponding author upon reasonable request.

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

## Acknowledgements

We thank the German Science Foundation (DFG) for financial support under grant No. SFB/TRR 80. S.B.-C. acknowledges IKERBASQUE and the Spanish Ministry of Economy, Industry and Competitiveness under the María de Maeztu Units of Excellence Program —MDM-2016-0618. The work at the University of California, Berkeley and Lawrence Berkeley National Laboratory was supported by the Office of Science, Office of Basic Energy Sciences, Materials Sciences and Engineering Division, of the U.S. Department of Energy (DOE) under Contract No. DE-AC02-05-CH11231 within the Quantum Materials Program (KC2202).

## Author contributions

G.C. and G.L. grew the samples and characterized them with help from M.B., D.P. and R. O.; M.B., A.F., E.S., M.M., E.d.S.N., and S.B.-C. conceived the experiments and performed them with the help of E.W., F.G., and H.S.; M.B. performed the data analysis with the help of A.F. and E.d.S.N.; E.d.S.N., M.M., S.B.-C., R.B., and B.K. supervised the project. M.B. and B.K. wrote the manuscript together with contributions from all authors.

## Additional information

**Competing interests:** The authors declare no competing interests.

