## [Peer Review File · Nature Communications]

Reviewers' comments:

Reviewer #1 (Remarks to the Author):

I have read the manuscript stabilization of 3D charge order in YBCO 6+x via epitaxial growth. By M. Bluschke and collaborators submitted to Nature Communications for consideration.

This manuscript reports the finding of 3D charge ordered CO state in the absence of magnetic field. Quasi 2D CO was previously observed in underdoped YBCO and shown to become 3D in high magnetic fields. The period deduced from quantum oscillations suggest an electron pocket resulting from the reconstruction (nesting) of the Fermi surface by the CO wave vector. The observation of a 3D CO state in the absence of magnetic field in this manuscript is novel and important as it may shed completely new light on the competing nature of CO and superconducting orders. Before recommending acceptance I would like to bring the following issues to author's attention.

1) The use of resonant x ray scattering at the Cu edge has enabled assessing the contribution of Cu crystallographic sites out of plane to the long range CO state. The manuscript conveys the message that it is precisely the chain Cu which couples the 2D CO and enables the 3D charge ordered state in the absence of magnetic field. I find intriguing that the 3D charge order disappears for the more deoxygenated sample with T_c of 41 K. Is this not suggesting that more than the Cu in the chains, the oxygen in the chains may be playing a role in the dimensionality of the charge ordered state? Especially since as proposed, the absence of CuO ortho-oxygen chain ordering (in the latter case presumably ortho-II) promotes nesting of the Fermi surface. On the other hand, the more deoxygenated are films the stronger is the 2D character of the superconductivity as shown previously in Phys. Rev. B 60, 15423 (1999). Is this dimensionality reduction also affecting the CO state? Could authors comment on this?

2) In previous studies on oxygen depleted YBCO single crystals, observing the 3D charge order required the application of high magnetic fields to suppress superconductivity. What are the implications of the observation of 3D order in superconducting films in the absence of magnetic field regarding the role or importance of this competition in the mechanism of the superconductivity?

In summary, this manuscript offers a set of interesting new data which could contribute to the understanding of the complex phenomena that meet under the superconducting dome of the cuprates. As such, my recommendation is that this manuscript should be published after the issues raised above have been settled. I can foresee that it will have strong impact on the condensed matter physics community.

Reviewer #2 (Remarks to the Author):

The authors report the observation of 3D charge density wave (CDW) order in epitaxially grown samples of YBCO. Similar to single crystals of YBCO which have been widely studied with x-ray diffraction, these thin films exhibit 2D CDW order. However, Bluschke et al. demonstrate that films grown on STO substrates, such that the resulting materials are tetragonal instead of orthorhombic, also exhibit 3D CDW order with relatively narrow Bragg peaks at an integer value of L . As noted by Bluschke, such phenomena has been observed previously, but only in high magnetic fields and only along the $(K\ 0\ L)$. Moreover, it remains a mystery why CDW order evolves into a more 3D version at high magnetic fields. While this study does not provide a complete answer to that problem, it provides a significant advance, shedding light on how CDW manifests in cuprates

crystal structure and symmetry are modified. Perhaps more generally, it highlights how tuning the structure of materials via epitaxial growth may be an open new directions to study CDW order in the cuprates. As such, I recommend publication of the paper in Nature Communications.

Prior to publication, I have several questions and suggestions that should be addressed prior to publication.

1. What is the c-axis correlation length of the (0.329 0 1) peak? How does it compare to with the (0 K 1) peak from high magnetic field measurements? It may be helpful to provide a plot of I vs. L through the 0.329 0 1) peak.
2. It should be specified how were the energy scans shown in fig. 3 performed. Are these energy scans at fixed Q? How was the background subtracted for these measurements?
3. The authors should provide numbers on the y axis of fig.'s 1 and 2 depicting the intensity vs Q, similar to fig. 3f.
4. The energy dependence of the (0.329 0 1) peak depicted in fig. 3 is intriguing. The authors make a case in the manuscript that the enhancement of the resonant scattering intensity at energies associated with transitions to states of both the Cu(I) and Cu(II) indicate both planes and chains are involved in the scattering. I find this interpretation compelling. However, since the d9 L states of the Cu(II) are also around the energy where the scattering is peaked, it may be possible that only a modulation of Cu (II) sites can also explain the energy dependence of the (0.329 0 1) peak. Achkar et al. (PRL 110, 017001 (2013)) showed that a modulation of the "valence" (ie the occupation of Cu(II) 3d9 L states) would lead to a peak in the scattering intensity ~ 1 eV above the 3d9 peak of the XAS, which may account for some of the scattering around 932.5 eV. I will leave it to the authors discretion whether to incorporate this alternate scenario into the manuscript.

We thank the referees for the careful reading of our manuscript, and for bringing important considerations to our attention. In the following we offer a point-by-point reply to their remarks and describe the modifications of the manuscript we made in response to their suggestions. We have also attached a version of the revised manuscript where the modifications are highlighted in yellow.

Reviewer #1 (Remarks to the Author):

I have read the manuscript stabilization of 3D charge order in YBCO 6+x via epitaxial growth. By M. Bluschke and collaborators submitted to Nature Communications for consideration.

This manuscript reports the finding of 3D charge ordered CO state in the absence of magnetic field. Quasi 2D CO was previously observed in underdoped YBCO and shown to become 3D in high magnetic fields. The period deduced from quantum oscillations suggest an electron pocket resulting from the reconstruction (nesting) of the Fermi surface by the CO wave vector. The observation of a 3D CO state in the absence of magnetic field in this manuscript is novel and important as it may shed completely new light on the competing nature of CO and superconducting orders. Before recommending acceptance I would like to bring the following issues to author's attention.

We thank the referee for the positive and constructive comments.

1) The use of resonant x ray scattering at the Cu edge has enabled assessing the contribution of Cu crystallographic sites out of plane to the long range CO state. The manuscript conveys the message that it is precisely the chain Cu which couples the 2D CO and enables the 3D charge ordered state in the absence of magnetic field. I find intriguing that the 3D charge order disappears for the more deoxygenated sample with T_c of 41 K. Is this not suggesting that more than the Cu in the chains, the oxygen in the chains may be playing a role in the dimensionality of the charge ordered state? Especially since as proposed, the absence of CuO ortho-oxygen chain ordering (in the latter case presumably ortho-II) promotes nesting of the Fermi surface. On the other hand, the more deoxygenated are films the stronger is the 2D character of the superconductivity as shown previously in Phys. Rev. B 60, 15423 (1999). Is this dimensionality reduction also affecting the CO state? Could authors comment on this?

The referee has correctly pointed out that not only the Cu(I) sites in the chain layers, but also the oxygens in the chain layers may play an important role in providing the necessary conditions for the stabilization of 3D-CO. While we maintain that communication of CO phase information in the c-direction is most likely achieved via the Cu(II)-O-Cu(I)-O-Cu(II) pathway (since this is the only bonding path along the c direction with significant covalency), the degree of oxygenation in the chain layer has direct impact on this pathway. As the oxygen content is decreased towards $\text{YBa}_2\text{Cu}_3\text{O}_6$ the Cu(I) valence tends towards the closed shell configuration $3d^{10}$. In this configuration, the Cu(I) ions are unable to conduct electrons between CuO_2 plane bilayers, thereby inhibiting c-axis communication of phase information associated with 2D-CO correlations, and effectively constraining the physics of the CuO_2 planes to two dimensions. We now mention this line of argumentation explicitly in the third paragraph of the discussion

section, and have added a citation [Ref. 33] to the work performed on YBCO thin films reported in *Phys. Rev. B* **60**, 15423 (1999).

We further note that the presence of 2D-CO correlations is expected to be a precondition for the stabilization of 3D-CO mediated by the Cu(I) sites. As such, the presence of 3D-CO is expected to be limited to a doping range in which both 2D-CO correlations are intrinsic, and in which sufficient charge carriers have been doped into the Cu(I) sites so as to realize active electronic Cu(II)-O-Cu(I)-O-Cu(II) pathways along the c-direction.

2) In previous studies on oxygen depleted YBCO single crystals, observing the 3D charge order required the application of high magnetic fields to suppress superconductivity. What are the implications of the observation of 3D order in superconducting films in the absence of magnetic field regarding the role or importance of this competition in the mechanism of the superconductivity?

The referee raises an important question regarding the competition between CO correlations and superconductivity. In previous studies on bulk YBCO the competing nature of the 2D-CO and superconductivity was evidenced in the form of a suppression of CO correlations below the superconducting critical temperature T_c [Ref. 2, 14, 27]. Further evidence for the competition between these two states was demonstrated in the presence of applied magnetic fields which simultaneously weaken superconductivity and enhance CO correlations [Ref. 14, 27]. For fields greater than 15 T a 3D-CO state is observed in bulk YBCO, which likewise competes with superconductivity, and coexists with a strengthened 2D-CO. In YBCO films grown on STO substrates, clear distinctions in the CO phenomenology are observed. Unlike in bulk crystals, both 2D- and 3D-COs are observed in zero field, and no temperature anomaly is observed at T_c in the intensity of the 3D-CO reflection. Furthermore, the 3D-CO state is stabilized at temperatures well above that of the 2D-CO state. We propose two possible scenarios to explain this phenomenology.

1. Mesoscopic phase separation between regions hosting 3D-CO and regions hosting 2D-CO + superconductivity. In this scenario the 3D-CO state does not exist homogeneously throughout the sample, but only in certain patches. Only in regions where 3D-CO is absent do 2D-CO correlations and superconductivity develop at low temperatures. As such only the 2D-CO correlations coexist locally with superconductivity, and experience a suppression below T_c .
2. Homogeneous coexistence of superconductivity and both 2D- and 3D-CO correlations. In this scenario the 3D-CO state is sufficiently well established at T_c , such that the onset of superconductivity does not weaken it. Rather, the superconductivity in our films may be suppressed by the presence of 3D CO, thereby resulting in reduced T_c compared to bulk crystals with the same oxygen content. In contrast the 2D-CO correlations are only weakly established at T_c and respond more dramatically to the onset of superconductivity.

Distinguishing between these two scenarios represents a great experimental challenge and would require the application of a spatially resolved technique capable of imaging CO domains. Therefore, we leave this for future studies. The two scenarios described above are now referred to in the discussion section at the end of the second paragraph.

In summary, this manuscript offers a set of interesting new data which could contribute to the understanding of the complex phenomena that meet under the superconducting dome of the cuprates. As such, my recommendation is that this manuscript should be published after the issues raised above have been settled. I can foresee that it will have strong impact on the condensed matter physics community.

Reviewer #2 (Remarks to the Author):

The authors report the observation of 3D charge density wave (CDW) order in epitaxially grown samples of YBCO. Similar to single crystals of YBCO which have been widely studied with x-ray diffraction, these thin films exhibit 2D CDW order. However, Bluschke et al. demonstrate that films grown on STO substrates, such that the resulting materials are tetragonal instead of orthorhombic, also exhibit 3D CDW order with relatively narrow Bragg peaks at an integer value of L . As noted by Bluschke, such phenomena has been observed previously, but only in high magnetic fields and only along the $(K\ 0\ L)$. Moreover, it remains a mystery why CDW order evolves into a more 3D version at high magnetic fields. While this study does not provide a complete answer to that problem, it provides a significant advance, shedding light on how CDW manifests in cuprates crystal structure and symmetry are modified. Perhaps more generally, it highlights how tuning the structure of materials via epitaxial growth may be an open new directions to study CDW order in the cuprates. As such, I recommend publication of the paper in Nature Communications.

We are grateful to the referee for the careful reading of our manuscript, and we believe that the insightful suggestions provided have helped to significantly strengthen our manuscript..

Prior to publication, I have several questions and suggestions that should be addressed prior to publication.

1. What is the c -axis correlation length of the $(0.329\ 0\ 1)$ peak? How does it compare to with the $(0\ K\ 1)$ peak from high magnetic field measurements? It may be helpful to provide a plot of I vs. L through the $(0.329\ 0\ 1)$ peak.

The c -axis correlation length of the $(0.329\ 0\ 1)$ peak is observed to be as high as 6 nm. Similar values have been reported for the field induced 3D-CO in bulk YBCO at low temperatures and in magnetic fields >17 T [Ref. 15, 16, 17]. An explicit comparison to the correlation length of the field-induced 3D-CO in bulk crystals is now provided in the second sentence of the section “ c -axis correlation and a - b anisotropy.” Inspired by the suggestion of the referee we have included cuts of I vs. L through the $(0.329\ 0\ 1)$ peak in Fig. 2d, and in Fig. 2e we show examples of

Lorentzian fitting to the background-subtracted data. The results of these fits are now listed alongside the analysis of the corresponding H -scans in Table 1.

2. It should be specified how were the energy scans shown in fig. 3 performed. Are these energy scans at fixed Q ? How was the background subtracted for these measurements?

Prompted by the referee's comment, we have added a paragraph to the methods section describing the details of our measurement and analysis.

3. The authors should provide numbers on the y axis of fig.'s 1 and 2 depicting the intensity vs Q , similar to fig. 3f.

This change has been implemented, and care has been taken that all intensities correspond directly to the normalized detector signal of the UE46 scattering chamber (up to a constant multiplicative factor of 10^7 , applied to all intensities). In order that the presentation of intensity absolute values be consistent, the intensities indicated in Fig. 3f and the peak areas quoted in Table 1 have been scaled accordingly. Only in the waterfall plots of Fig. 2c-e some of the curves were shifted for clarity.

4. The energy dependence of the $(0.329\ 0\ 1)$ peak depicted in fig. 3 is intriguing. The authors make a case in the manuscript that the enhancement of the resonant scattering intensity at energies associated with transitions to states of both the Cu(I) and Cu(II) indicate both planes and chains are involved in the scattering. I find this interpretation compelling. However, since the $d9\ L$ states of the Cu(II) are also around the energy where the scattering is peaked, it may be possible that only a modulation of Cu (II) sites can also explain the energy dependence of the $(0.329\ 0\ 1)$ peak. Achkar et al. (PRL 110, 017001 (2013)) showed that a modulation of the "valence" (ie the occupation of Cu(II) $3d9\ L$ states) would lead to a peak in the scattering intensity ~ 1 eV above the $3d9$ peak of the XAS, which may account for some of the scattering around 932.5 eV. I will leave it to the authors discretion whether to incorporate this alternate scenario into the manuscript.

The referee's comment is indeed very insightful, and we admit to having considered at length the very same possibility. We have found two arguments which have finally led us to exclude a pure valence modulation in the CuO₂ plane (Cu(II) sites), and instead settle for the interpretation presented in the manuscript.

1. A careful scrutiny of the Cu L_3 resonant scattering lineshapes in Fig. A reveals a significantly smaller width of the resonance associated with the Cu(II) valence modulation model compared to the resonance of the $(0.329\ 0\ 1)$ peak presented in our manuscript. Furthermore, in the valence modulation model there is almost no intensity directly at the energy corresponding to the XAS maximum, whereas for the $(0.329\ 0\ 1)$ peak, the scattering resonance is almost maximized at the energy of the XAS maximum.

Figure A: Comparison of the energy dependent resonant scattering from the (0.329 0 1) peak in a YBCO film grown on STO, and the Cu(II) valence modulation model published in Achkar *et al.* Phys. Rev. Lett. **110**, 017001 (2013). Both the valence modulation model and the XAS data from Achkar *et al.* are plotted on an energy scale which is shifted by 0.18 eV. The energy shift was chosen so as to align the main XAS peak in the two studies.

2. The YBCO films investigated in this study are closely similar to bulk YBCO crystals. Structural X-ray diffraction indicates lattice parameters consistent with oxygen-disordered bulk YBCO, transport and magnetization measurements indicate superconducting transitions and residual resistivities comparable to bulk samples, and measurements of the 2D-CO rod indicate charge correlations which are suppressed below T_c and whose in-plane correlation lengths are comparable to those observed in bulk underdoped YBCO. The most significant difference between our YBCO films grown on STO and previously studied detwinned, underdoped bulk YBCO crystals is the lack of CuO chain ordering in the charge reservoir layer. As discussed in the text, this lack of CuO chain order leads to an effectively tetragonal structure, which is expected to strengthen the Fermi surface nesting responsible for CO correlations. In contrast, we were unable to identify any possible mechanism linking the absence of oxygen order and the 3D-CO in the 'valence modulation' model. As such we identify the most natural interpretation to be that in which the additional resonant intensity approximately 1-2 eV above the main XAS maximum is associated with Cu(I) states participating in the 3D CO.

In light of these two lines of argumentation, we have chosen to omit from the text an explicit reference to the Cu(II) valence modulation model.

REVIEWERS' COMMENTS:

Reviewer #1 (Remarks to the Author):

I have read the revised manuscript "Stabilization of 3D charge order in YBCO $6+x$ via epitaxial growth" by M. Bluschke and collaborators and author's response to the first review round.

Authors have satisfactorily addressed the issues raised in my previous report concerning 1) the role of reduced dimensionality of the superconductivity of the cuprates upon reduction of the oxygen content and 2) the competing nature of the 3D CO- and the superconducting states. Changes in the manuscript have been made according to this criticism.

I maintain my view on the importance of the finding of a 3D charge ordered CO state in underdoped cuprate films (in the absence of magnetic field) reported in this manuscript, as it may help shedding light on the role of competing orders in the mechanism of the High T_c Superconductivity. My recommendation is to publish this manuscript in Nature Communications.

Reviewer #2 (Remarks to the Author):

The authors have adequately addressed the concerns of my initial review. Accordingly, I recommend the paper for publication in Nature Communications.